# Boosting Offline Reinforcement Learning via Data Rebalancing

Yang Yue [1 2] [*] Bingyi Kang [1] [†] Xiao Ma [1] Zhongwen Xu [1] Gao Huang [2] Shuicheng Yan [1]

## Abstract

Offline reinforcement learning (RL) is challenged by the distributional shift between learning policies and datasets. To address this problem, existing works mainly focus on designing sophisticated algorithms to explicitly or implicitly constrain the learned policy to be close to the behavior policy. The constraint applies not only to well-performing actions but also to inferior ones, which limits the performance upper bound of the learned policy. Instead of aligning the densities of two distributions, aligning the supports gives a relaxed constraint while still being able to avoid out-of-distribution actions. Therefore, we propose a simple yet effective method to boost offline RL algorithms based on the observation that resampling a dataset keeps the distribution support unchanged. More specifically, we construct a better behavior policy by resampling each transition in an old dataset according to its episodic return. We dub our method ReD (Return-based Data Rebalance), which can be implemented with less than 10 lines of code change and adds negligible running time. Extensive experiments demonstrate that ReD is effective at boosting offline RL performance and orthogonal to decoupling strategies in long-tailed classification. New state-of-the-arts are achieved on the D4RL benchmark.

## 1. Introduction

Recent advances in Deep Reinforcement Learning (DRL) have achieved great success in various challenging decision-making applications, such as board games (Schrittwieser et al., 2020) and strategy games (Vinyals et al., 2019). However, DRL naturally works in an online paradigm where agents need to actively interact with environments for experience collection. This hinders DRL from applications in

[1]Sea AI Lab [2]Department of Automation, BNRist, Tsinghua University. Correspondence to: Yang Yue <le-y22@mails.tsinghua.edu.cn>. [*]This work was done when Yang Yue was an intern at Sea AI Lab. [†]Corresponding Author.

the real-world scenarios where interactions are prohibitively expensive and dangerous. Offline Reinforcement learning attempts to address the problem by learning from previously collected data, which allows utilizing large datasets to train agents (Lange et al., 2012). Vanilla off-policy RL algorithms suffer poor performance in the offline setting due to the distributional shift problem (Fujimoto et al., 2019). Specifically, performing policy evaluation, i.e., updating value function with Bellman's equations, involves querying the value of out-of-distribution (OOD) state-action pairs, which potentially leads to accumulative extrapolation error. The main class of existing methods alleviates the above problem via constraining the learned policy not to deviate far from the behavior policy by ***directly restricting their probability densities***. The constraint can be KL divergence (Jaques et al., 2019; Peng et al., 2019), Wasserstein distance (Wu et al., 2019), maximum mean discrepancy (MMD) (Kumar et al., 2019), or behavior cloning regularization (Fujimoto & Gu, 2021).

However, such constraints might be too restrictive as the learned policy is forced to mimic both bad and good actions of the behavior policy. For instance, consider a dataset $\mathcal{D}$ for state space $\mathcal{S}$ and action space $\mathcal{A} = \{a_1, a_2, a_3\}$ collected with behavior policy $\beta$. At one specific state $s^*$, the policy $\beta$ assign probability 0.2 to action $a_1$, 0.8 to $a_2$ and zero density to $a_3$. However, $a_1$ would lead to much higher expected return than $a_2$. Minimizing the density distance of two policies can avoid $a_3$, but forces the learned policy to choose $a_2$ over $a_1$, resulting in much worse performance. Therefore, a more reasonable condition is to constrain two policy distributions to have the same support of action, *i.e.*, the learned policy has positive density only on actions that give non-zero probability in the behavior policy (Kumar et al., 2019). In this case, it regularizes the learning policy to sample in-distribution state-action pairs and gives a higher performance upper bound. We term it ***support alignment*** as a more flexible relaxation of the behavior regularization. Nevertheless, explicit support alignment is intractable in practice (Kumar et al., 2019), especially for high-dimensional continuous action spaces.

We make one important observation that reweighting the data distribution density does not change the support of the data distribution, *i.e.*, zero density is still zero density after reweighting. In the context of offline RL, instead of

sampling uniformly from the offline dataset, varying the sampling rate of offline samples to focus on trajectories with higher accumulative returns, *i.e.*, changing the action density, does not change the support and produces a better behavior policy. Matching the learned policy with a resampled policy is approximately performing support alignment.

In this work, we boost offline RL by designing data rebalancing strategies to construct better behavior policies. We first show that existing offline datasets are extremely imbalanced in terms of episodic return (as shown in Fig. 1). In some datasets, most actions lead to a low return, which renders the possibility that current density-based constraints are too restrictive. We thus propose to resample the dataset during training based on episodic return, which assigns larger weights to transitions with higher returns. The method is thus dubbed Return-based Data Rebalance (ReD), and can be easily implemented with less than 10 lines of code. Without any modification to prior hyperparameters, we find that ReD effectively boosts the performance of various popular offline RL algorithms by a large margin on diverse domains in D4RL (Brockman et al., 2016; Fu et al., 2020). Then as our minor contribution, we propose a more elaborated implementation of data rebalance, Decoupled ReD (DeReD), inspired by decoupling strategies for data rebalance training in long-tailed classification (Kang et al., 2020). The proposed DeReD combined with IQL achieves the state-of-the-art performance on D4RL. The effectiveness of return-based data rebalance may imply that data dimension is as important as algorithmic dimension in offline RL.

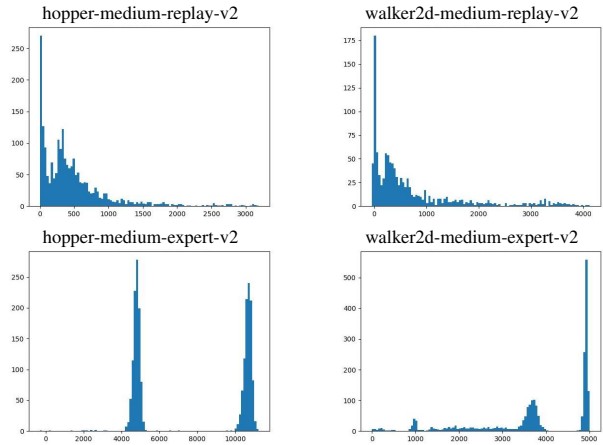

*Figure 1.* Visualization of Trajectory Return Distributions. Medium-replay datasets are likely to have a long-tailed distribution, and medium-expert are likely to have two peaks.

## 2. Related Work

**Offline RL.** To alleviate extrapolation error and address the distributional shift problem, a general framework for prior offline RL works is to constrain the learned policy to stay close to the behavior policy. Considering KL-divergence is easy to accurately compute under the Gaussian distribution assumption, many works choose KL-divergence as policy constraint. There are many concrete implementation choices, *e.g.*, explicitly modeling behavior prior by VAE, avoiding explicit modeling by the dual form (Wu et al., 2019; Jaques et al., 2019). Exponentially advantage-weighted regression, an implicit form of KL-divergence constraint, is derived by AWR (Peng et al., 2019), CRR (Wang et al., 2020) and AWAC (Nair et al., 2020). IQL (Kostrikov et al., 2021b) also extracts policy via advantage-weighted regression from the expectile value function, enforcing a KL constraint. Behavior cloning (BC) is another alternative to implement constraint (Fujimoto & Gu, 2021). There also exist other directions to realize the offline RL. One line is to regularize Q-function by conservative estimate (Kumar et al., 2020; Buckman et al., 2020). Surprisingly, our experiments show that data rebalance also works with conservative Q-learning (Kumar et al., 2020). Another line is to view offline RL as the sequential modeling problem by masked transformer (Chen et al., 2021; Janner et al., 2021), and then the transformer outputs actions to attain the given return.

Some works attempt to approximately satisfy support alignment. BEAR (Kumar et al., 2019) utilizes maximum mean discrepancy to approximately optimize support alignment. However, the effectiveness that the sampled MMD has in constraining two distributions to the same support is only empirically shown in a low-dimension distribution with diagonal covariance matrices with no theoretical guarantee, and MMD is extremely complex to implement. That's a possible reason why Wu et al. finds MMD has no gain compared to KL. Another attempt to relax the restrictive constraint is adaptively adjusting the weight of the constraint term by dual gradient ascent. However, Wu et al. observes that the adaptive weight has a modest *disadvantage* over a fixed one.

**Rebalance Data.** Dataset rebalance is widely used in visual tasks when facing a long-tailed distribution (Zhang et al., 2021). For decision making, imitation learning (IL) aims to learn from demonstration, where rebalance is naturally applied to filter out bad demonstrations. BAIL (Chen et al., 2020) employs a neural network to approximate the upper envelope (*i.e.*, the optimal return from data) and select good state-action pairs to imitate. Another rebalance in IL is 10%BC where behavior cloning only uses the top 10% of transitions ordered by episode return (Chen et al., 2021). MARWIL (Wang et al., 2018) and AWR (Peng et al., 2019) employ exponentially advantage-weighted behavior cloning, equivalent to policy improvement step with KL constraint in RL. Experiments show that our method can further improve KL constraint methods.

Table 1. Averaged normalized scores on MuJoCo locomotion tasks. we report the average over the final 10 evaluations and 5 seeds. We focus on discrepancy between "uniform" and "ReD". The results that have significant advantage over another are printed in bold type.

| | CRR | | CQL | | IQL | | TD3+BC | |
|---|---|---|---|---|---|---|---|---|
| | uniform | ReD | uniform | ReD | uniform | ReD | uniform | ReD |
| halfcheetah-medium-v2 | 42.2 | 43 | 48.1 | 48.2 | 47.6 | 47.6 | 48.2 | 48.5 |
| hopper-medium-v2 | 49.6 | 51 | 71.8 | 69.4 | 64.3 | 66 | 58.8 | 59.3 |
| walker2d-medium-v2 | **62.4** | 39 | 83.3 | 83.5 | 79.9 | 78.6 | 84.3 | 83.7 |
| halfcheetah-medium-replay-v2 | 37.7 | 38 | 45.2 | 46.3 | 43.4 | 44.3 | 44.6 | 44.7 |
| hopper-medium-replay-v2 | 21.5 | **66.4** | 95.3 | **98.6** | 89.1 | **101** | 58.1 | **77.4** |
| walker2d-medium-replay-v2 | 11.6 | **30.6** | 82.3 | **86.7** | 69.6 | **79.5** | 73.6 | **82.3** |
| halfcheetah-medium-expert-v2 | **79.5** | 55 | 66.2 | **81.6** | 83.5 | **92.6** | 93 | 93.2 |
| hopper-medium-expert-v2 | 54.1 | 56.6 | 76.9 | **95** | 96.1 | **106.1** | 98.8 | **106.2** |
| walker2d-medium-expert-v2 | 1 | **25.2** | 110 | 110 | 109.2 | 110.5 | 110.3 | 110 |
| mujoco-v2 total | 359.6 | **404.8** | 679.1 | **719.3** | 682.7 | **726.2** | 669.7 | **705.3** |

In online reinforcement learning, PER (Schaul et al., 2016) dynamically prioritizes experiences with a larger temporal-difference error. SIL (Oh et al., 2018) favors transitions based on episode return. To the best of our knowledge, few works apply data rebalance to offline RL (not including imitation learning). One work is RBS (Shen et al., 2021), which also focuses on data perspective. However, RBS designs more sophisticated strategies like upper envelope to dynamically modify the sampling weights, while our method adopts more simple and efficient static rebalance before training. Moreover, RBS is only tested with toy examples such as PyBullet Gymperium, while our work experiments with broad algorithms and much more complicated environments. Yarats et al. also reveals data generation is important for offline RL, but it focuses on how to collect exploratory data with unsupervised RL.

## 3. Return-based Data Rebalance

We now describe an efficient and effective approach to re-balancing the offline dataset and leading to a better behavior policy. Considering most RL algorithms work with continuous vector or pixel input and a state usually appears once in the dataset, it's inconvenient to rebalance dataset by directly changing the frequency of actions for a state. Instead, We rebalance the dataset by resampling as long-tailed training does (Zhang et al., 2021). For the transition $i$ in offline dataset $\mathcal{D}$ with size $N$, it should be uniformly sampled by the probability $P(i) = \frac{1}{N}$ in the prior offline RL paradigm, corresponding to the behavior policy $\beta$. After rebalance, the probability of sampling transition $i$ should be

$$P(i) = \frac{p_i^\alpha}{\sum_{k=1}^N p_k^\alpha}, \quad (1)$$

where $p_i$ is the normalized return of transition $i$. The exponent $\alpha \in [0, \infty)$, deciding the rebalance extent. $\alpha = 0$

corresponds to the uniform case, while $\alpha \to \infty$ means only sampling transitions in the best trajectory. The normalized return $p_i$ is computed by:

$$p(i) = \frac{R_i - R_{min}}{R_{max} - R_{min}} + p_{base}, \quad (2)$$

where $R_i$ is the episode return of the trajectory which the transition $i$ belongs to. $R_{min}$ and $R_{max}$ represents the minimal and maximal value in all trajectory returns. $p_{base}$ is zero or a small positive constant that prevents the marginal transitions not being visited. The definition ensures that the probability of sampling transition is monotonic in its trajectory return, roughly assigning larger probability to better transitions. Such a resampling is corresponding to a different behavior policy $\beta'$ of the same support which prefers actions that generate high returns. Intuitively, $\beta'$ is very likely to be better than $\beta$.

**implementation.** Our rebalance method is very easy to implement, just computing the trajectory returns and generating the sampling probability distribution $P$ before training begins. Then resample can be done by numpy.random.choice with $P$. These two changes cause very little computational overhead. Hyperparameters are simply set $\alpha = 1$ and $p_{base} = 0$, except for antmaze environments where $p_{based} = 0.2$ is used, because trajectory return in antmaze can be only 0 or 1. If $p_{base} = 0$, all trajectories with return 0 would be discarded.

## 4. Experiments

In this section, experiments on D4RL benchmark (Fu et al., 2020) are conducted to empirically show ReD can improve popular offline RL algorithms on diverse domains, demonstrating the effectiveness of data rebalance. Then we evaluate the proposed decoupling strategy DeReD on the D4RL

*Table 2.* Full Results and runtime of ReD and DeReD based on IQL on D4RL. Averaged normalized scores over the final 10 evaluations and 5 seeds on D4RL tasks except for Antmaze. we report the final evaluation and 5 seeds for Antmaze. Following IQL, we use "v2" environments for mujoco tasks and "v0" for other tasks. For DeReD, we use upward arrow to denote stage 2 has improvement over stage 1. The best score in every task is printed in bold type.

| | IQL | ReD | DeReD | |
| --- | --- | --- | --- | --- |
| | | | stage1 | stage2 |
| halfcheetah-medium | 47.6 | 47.6 | 47.5 | 47.6 |
| hopper-medium | 64.3 | **66** | 65.5 | 65.1 |
| walker2d-medium | 79.9 | 78.6 | 74.5 | **81.9**↑ |
| halfcheetah-medium-replay | 43.4 | **44.3** | 43.9 | 43.4 |
| hopper-medium-replay | 89.1 | **101** | 92.8 | 100.1↑ |
| walker2d-medium-replay | 69.6 | **79.5** | 72.9 | 77↑ |
| halfcheetah-medium-expert | 83.5 | 92.6 | 87.9 | **91.8**↑ |
| hopper-medium-expert | 96.1 | **106.1** | 89.3 | 104.7↑ |
| walker2d-medium-expert | 109.2 | **110.5** | 110.1 | **110.5** |
| mujoco total | 682.7 | **726.2** | 684.4 | 722.1↑ |
| antmaze-umaze | 88 | **89.2** | 84.8 | **89.6** |
| antmaze-umaze-diverse | 64 | **79.8** | 67.0 | 72.3↑ |
| antmaze-medium-play | 71.5 | **78.4** | 68.8 | 71.8 |
| antmaze-medium-diverse | 57.5 | 68.4 | 67.7 | **77.6**↑ |
| antmaze-large-play | 42.3 | 14.6 | 32.6 | **49.2**↑ |
| antmaze-large-diverse | 44 | 37.6 | 47 | **48.2** |
| antmaze total | 367.3 | 368 | 367.9 | **408.7**↑ |
| kitchen-complete-v0 | 66.3 | 62.7 | **67.4** | 64 |
| kitchen-partial-v0 | 53.3 | **69.5** | 47.6 | 66↑ |
| kitchen-mixed-v0 | 49.5 | 49.9 | **50.9** | 49.5 |
| kitchen-v0 total | 169.1 | **182.1** | 165.9 | 179.5↑ |
| pen-human-v0 | 74.8 | 83 | 76 | **88**↑ |
| hammer-human-v0 | 1.7 | 1.8 | 1.5 | **2.1** |
| door-human-v0 | 4.3 | 4.4 | 6.1 | **7.1** |
| relocate-human-v0 | 0.1 | 0.1 | 0.1 | 0.1 |
| pen-cloned-v0 | 40.9 | **66.6** | 37.8 | 53↑ |
| hammer-cloned-v0 | 1.8 | 1.1 | 1.6 | 1.8 |
| door-cloned-v0 | 1.8 | **4.9** | 2.4 | 4.4 |
| relocate-cloned-v0 | -0.2 | -0.2 | -0.2 | -0.2 |
| adroit-v0 total | 125.2 | **161.7** | 125.3 | 156.3↑ |
| total | 1344.3 | 1438.0 | 1351.8 | **1466.6** |
| runtime | 28min | 29min | 40min | |

benchmark. The result shows data rebalance can achieve the new state-of-the-art on D4RL.

We choose IQL as our baseline, because IQL (Kostrikov et al., 2021b) achieves the best performance on D4RL. We also choose diverse types of algorithms including CQL (Kumar et al., 2020), CRR (Wang et al., 2020), and TD3+BC (Fujimoto & Gu, 2021) to validate our rebalance method. Both IQL and CRR use advantage-weighted regression for policy improvement, namely KL divergence, while TD3+BC uses a direct behavior cloning term. CQL can also be derived by KL divergence between behavior policy and Boltzmann policy (Kostrikov et al., 2021a). To ensure a fair comparison, we first reproduce baselines' results. For IQL and TD3+BC, we rerun the author-provided code[1][2];

[1]https://github.com/ikostrikov/implicit_q_learning

[2]https://github.com/sfujim/TD3_BC

for CQL, we rerun a reimplementation JAX code[3], causing a slight discrepancy with PyTorch version results reported in the CQL paper. The CRR paper only tests on RL Unplugged (Gulcehre et al., 2020), so we reimplement CRR on D4RL by JAX. We run every algorithm with 1M gradient steps and evaluate every 5000 steps, except evaluating Antmaze every 100K steps. Each evaluation consists of 10 episodes. Following TD3+BC (Fujimoto & Gu, 2021), we report the average over the final 10 evaluations and 5 seeds except for Antmaze, for which we report the average over the final evaluation and 5 seeds. See "uniform" columns in Table 1 for reproduce results.

### 4.1. Results of ReD

To demonstrate simplicity and generality, we avoid changing any hyperparameters in baseline algorithms when running our rebalance method. "ReD" columns in Table 1 demonstrates the results of baseline algorithms trained by the weighted sampler described in Eqn. 1 and Eqn. 2. We found ReD can markedly improve the performance of algorithms, even though CQL and IQL have achieved quite high scores on mujoco tasks. We notice that the boost of ReD mainly comes from "medium-replay" and "medium-expert" level tasks. This observation adheres to our intuition of rebalancing dataset and deriving a better behavior policy. Fig. 1 demonstrates these datasets collected by a mixture of policies have trajectories with diverse quality and therefore have more potential to improve the behavior policy by data rebalance. Full visualization results of D4RL datasets are shown in Fig. 2. Actually, ReD can boost the performance of IQL in 3 out of 6 tasks, *i.e.*, umaze-diverse, medium-play, and medium-diverse. However, ReD has a dramatic drop in large-play and a slight drop in large-diverse. As for Kitchen and Adroit tasks, ReD also boosts IQL. It is noteworthy that ReD boosts the performance on kitchen-partial and pen-clone by around 50%, which may benefit from the return distributions with a small amount of high-return trajectories (see Fig. 2). Meanwhile, the last row of Table 2 demonstrates ReD only adds 1 minute time cost for resampling. Overall, ReD outperforms vallina IQL by a large margin and adds negligible runtime.

### 4.2. Decoupled Return-based Data Rebalance

Although ReD theoretically produces a higher accumulative return and empirically gives a strong performance on standard gym-locomotion tasks with dense rewards, it tends to hinder the performance on tasks with long-delayed sparse rewards, e.g., on antmaze-large environments. The reason behind is that on antmaze-large environments, all succeeded episodes have a reward of 1, while others have a reward of 0. ReD further reduces the importance of failed trajectories,

[3]https://github.com/young-geng/JaxCQL

without which an agent might fail to learn distinguishable values through Bellman's equation.

To tackle this issue, we darw insights from the long-tailed learning community (Kang et al., 2020) and we introduce Decoupled ReD (DeReD), a two-stage training framework for long-horizon delayed sparse reward scenarios. At the first stage, we train the agent with a uniform sampler to encourage the agent to learn meaningful value estimation and policies. Next, we freeze the policy head and value head of the network, and finetune the feature extraction backbone with ReD to further improve the agent with a higher return bound. As a result, DeReD significantly outperforms baseline methods on the antmaze-large environments. Although, DeReD still improves the baseline IQL on smaller domains on antmaze-uname and antmaze-medium, it shows a negative effect when compared with ReD. This is potentially because on domains with a shorter horizon, distinguishing states during policy evaluation is less difficult and freezing the policy / value head trained by uniform samplers limits its capability of ReD on these tasks. More details and experiments about DeReD can be found at Appendix A.1.

### 4.3. The influence of Hyperparameter $p_{base}$

*Table 3.* Effect of hyperparamter $p_{base}$.

| $p_{base}$ | 0 | 0.2 | 0.5 | 1.0 | $\infty$ (IQL) |
|---|---|---|---|---|---|
| halfcheetah-medium | 47.6 | - | 47.4 | - | 47.6 |
| hopper-medium | 66 | - | 64.1 | - | 64.3 |
| walker2d-medium | 78.6 | - | 78.9 | - | 79.9 |
| halfcheetah-medium-replay | 44.3 | - | 44.3 | - | 43.4 |
| hopper-medium-replay | **101** | - | 99.2 | - | 89.1 |
| walker2d-medium-replay | **79.5** | - | 61.3 | - | 69.6 |
| halfcheetah-medium-expert | **92.6** | - | 90.6 | - | 83.5 |
| hopper-medium-expert | **106.1** | - | 84.9 | - | 96.1 |
| walker2d-medium-expert | 110.5 | - | 110.1 | - | 109.2 |
| mujoco total | **726.2** | - | 680.8 | - | 682.7 |
| antmaze-umaze | 86.8 | 89.2 | 85.6 | **91.4** | 88 |
| antmaze-umaze-diverse | 37 | **79.8** | 65.8 | 60.4 | 64 |
| antmaze-medium-play | 37.6 | **78.4** | 72.8 | 72.2 | 71.5 |
| antmaze-medium-diverse | 32.8 | 68.4 | 64 | **74.2** | 57.5 |
| antmaze-large-play | 23.4 | 14.6 | 20.6 | 24.4 | **42.3** |
| antmaze-large-diverse | 24.4 | 37.6 | 42.4 | **44.4** | 44 |
| antmaze total | 242 | **368** | 351.2 | 367 | **367.3** |

In Table 3, We also demonstrates the effect of hyperparameter $p_{base}$ in Eqn. 2 on ReD. With larger $p_{base}$, the probability of sampling transition with different returns would be closer, and when $p_{base}$ goes to $\infty$, ReD degenerates to uniform sample. For mujoco, we find when $p_{base}$ equals 0.5, the effect of ReD has decreased to the performance of IQL. Antmaze's return can be only zero or one (see Fig. 2). $p_{base} = 0$ causes the trajectories where the agent don't reach the target have no chance to be sampled, resulting in a catastrophic drop. When adopting a milder resample, *i.e.*, $p_{base}$ equals to 0.2, 0.5, 1.0, ReD boosts the performance on 'antmaze-umaze' and 'antmaze-medium' but drops in antmaze-large-play compared to IQL ($p_{base} \to \infty$). When

$p_{base}$ goes larger, the boost and drop both approach a smaller value.

### 4.4. Compare Different Data Rebalance Methods

*Table 4.* Analyze different rebalance methods based on CQL.

| | CQL | return resample | CQL[10%] | reward resample |
|---|---|---|---|---|
| halfcheetah-medium | 48.1 | 48.2 | 47 | 48.2 |
| hopper-medium | 71.8 | 69.4 | 65.5 | 68.2 |
| walker2d-medium | 83.3 | 83.5 | 76.5 | 83 |
| halfcheetah-medium-replay | 45.2 | 46.3 | 41.8 | 46.4 |
| hopper-medium-replay | 95.3 | **98.6** | 96.8 | 97.7 |
| walker2d-medium-replay | 82.3 | **86.7** | 67.9 | 83.9 |
| halfcheetah-medium-expert | 66.2 | **81.6** | 73.3 | 74.2 |
| hopper-medium-expert | 76.9 | 95 | **106.6** | 87 |
| walker2d-medium-expert | 110 | 110 | 109.3 | 109.4 |
| mujoco total | 679.1 | **719.3** | 684.7 | 698 |

In this section, we compare different data rebalance methods along two dimensions - what should be rebalanced and how to implement rebalance. we choose CQL as our baseline. For the former question, we conduct experiments to rebalance the dataset by return and reward, denoted as **return-resample** and **reward-resample**. Table 4 shows reward rebalance boosts vallina CQL but falls behind return rebalance, revealing that trajectory return is a better indicator than transition reward to evaluate behavior quality. For the latter question, we test an invariant where the agent is trained in top 10% transitions sorted by trajectory return, denoted as **CQL[10%]**. It achieves the best score in hopper-medium-expert, but performs not well in many tasks.

## 5. Conclusion

We present a training pipeline to boost offline RL algorithms by return-based data rebalance. ReD adds nearly no computation burden and can be easily implemented. Despite its simplicity and efficiency, experiments show that it can significantly improve the performance of prior popular algorithms. Then we propose DeReD by decoupling training into two stages, which combined with IQL achieves the state-of-the-art on the D4RL benchmark. The effectiveness of data rebalance may indicate data rebalance is a promising research direction for offline RL.

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

# Appendix

## A. Additional Experiment Results

### A.1. DeReD Details

At the first stage, we train the agent with 1M gradient steps. Then the agent is finetuned with 200K steps, except mujoco where we finutune with 500K steps. Longer gradient steps will not cause fairness concerns, because usually 1M steps are suitable for offline algorithms in D4RL and more steps will impair the performance due to overfitting. We don't change any hyperparameters in prior algorithms. Actor and critic are both three-layer MLP in IQL. During the second stage, we only finetune the first two layers with 0.1x learning rate and fix the last one, which performs a little bit better than finetuning all layers denoted as **DeReD-A** (see Table 5). It may be because data rebalance mainly helps learn representation, rather than value prediction.

*Table 5.* Compare DeReD and DeReD-A on D4RL.

|                 | DeReD  | DeReD-A |
| --------------- | ------ | ------- |
| mujoco total    | 722.1  | 715.5   |
| antmaze total   | 408.7  | 390.6   |
| kitchen-v0 total| 179.5  | 178.2   |
| adroit-v0 total | 156.3  | 146     |
| total           | 1466.6 | 1430    |

## B. Imbalance Visualization

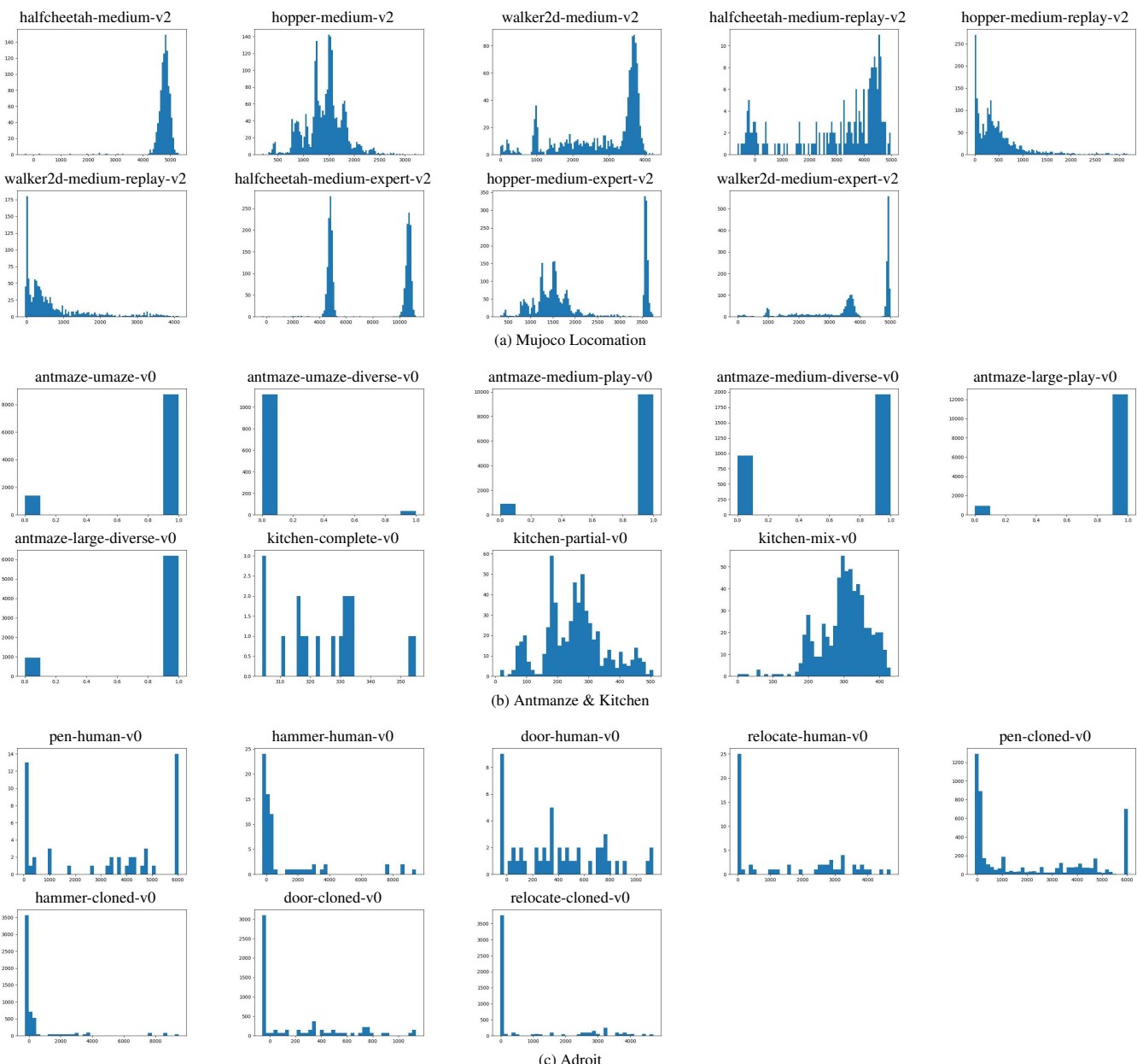

*Figure 2.* Full Visualization of Trajectory Return Distributions.