# OpenReview forum: "Boosting Offline Reinforcement Learning via Data Rebalancing"
_NeurIPS.cc/2022/Workshop/Offline_RL — Offline RL Workshop NeurIPS 2022_

### Official Review · Reviewer_f7Sv · 2022-10-07
**A simple idea with limited new results.**

**Rating:** 5
**Confidence:** 5

**Review:**

This work introduced the Return-based Data Rebalance (ReD) to resample offline datasets based on episodic returns, where we assign larger weights to transitions with higher returns. The key idea is to focus on trajectories with higher accumulative returns. For example, using the cumulative trajectory return as the sample weight in PER.

The main idea is similar to BAIL (Chen et al., 2020), where we first use a metric, i.e., cumulative return, to select a subset of high-quality samples, and then assign higher weights to these samples to learn an offline RL agent. The result is not surprising. Just like agents learned from the `-expert-` dataset performs better than agents learned from the `-medium-` dataset, we sometimes can improve the performance of offline RL agents by down-weighting low-quality samples (with lower cumulative returns).

Pros:
- The paper is well-written and easy to follow.
- Experiments on the D4RL benchmark show that ReD can achieve better performance.
- This paper introduced DeReD to further improve ReD.

Cons:
- The main weakness is the limited novelty, and the results are not surprising.
- Performance gain in the `-medium-v2` level dataset is not significant.
- It requires the offline dataset to contain full trajectories. However, in some real-world applications, we might only have transition pairs instead of a full trajectory.
- The effectiveness of ReD in some tasks with sparse rewards is doubtful, i.e., `ant-maze-large`. Because many  episode return $R_i$ has the same value $1$ or $0$, which makes ReD very close to uniform sampling. Though this paper introduced DeReD to mitigate this problem, it was also more computationally expensive.

---

### Official Review · Reviewer_ctfZ · 2022-10-18

**Rating:** 5
**Confidence:** 5

**Review:**

This paper proposes a new paradigm to improve behavioral cloning with (re)sampling of trajectories with higher returns. To accomplish this, the proposed method first pre-computes the trajectory returns and then calculates the sampling probability distribution of each trajectory. During training, these probabilities are used to sample data. To show the effectiveness of this method,  they run experiments on the D4RL benchmark where it is added on top of IQL, CQL, and CRR.

While this method shows some improvement, it is not clear if these results are statistically significant as standard deviation is not reported. In addition, this paper assumes that trajectory returns can be computed as it might not always be the case ( imagine batch data are shuffled). Finally, it is not clear to me why Antmaze's settings should be different.  Finally, it would be interesting to see how this method helps if it be used with methods like BC and BCQ.

Other comments:
-- adding algorithm blocks can be useful.
-- In line 60 "MMD is extremely complex to implement." I don't think that is true.
-- related work about "Offline RL" needs revision as a couple of works regarding regularizing Q-function and model based methods are missing.